# Dynamic Micro-Vibration Measurement Based on Orbital Angular Momentum

**Jialong Zhu** [1,2] , **Fucheng Zou** [3,*], **Le Wang** [2] **and Shengmei Zhao** [2,4,*]

1    School of Information Engineering, Suqian University, Suqian 223800, China
2    Institute of Signal Processing and Transmission, Nanjing University of Posts and Telecommunications (NUPT), Nanjing 210003, China
3    School of Electronic Information and Automation, Aba Teachers University, Wenchuan 623002, China
4    National Laboratory of Solid State Microstructures, Nanjing University, Nanjing 210093, China
*    Correspondence: 20189605@abtu.edu.cn (F.Z.); zhaosm@njupt.edu.cn (S.Z.)

**Abstract:** In this study, we introduce a novel approach for dynamic micro-vibration measurement, employing an Orbital Angular Momentum (OAM) interferometer, where the reference beam is Gaussian (GS) and the measurement beam is OAM. The OAM light reflected back from the target carries information about the target's vibrations. The interference of the OAM light with Gaussian light generates petal-shaped patterns, and the target's vibration information can be measured by detecting the rotation angle of these petals. Our proposed method demonstrates enhanced tolerance to misalignment and superior precision. The effects of vibration frequency, CCD frame rates, and Topological Charges (TCs) on measurement accuracy are analyzed thoroughly. Experimental results reveal that the proposed method offers a higher accuracy (up to 22.34 nm) and an extended measurement range of (0–800 cm). These capabilities render our technique highly suitable for applications demanding nanometer-scale resolution in various fields, including precision engineering and advanced optical systems.

**Keywords:** orbital angular momentum; micro-vibration measurement; dynamic measurement; unidimensional vibration measurement; nanometer-scale precision





## 1. Introduction

Vibration, a ubiquitous phenomenon stemming from mechanical oscillations, is an integral aspect of our environment. Its measurement is critically important across various sectors, playing a key role in monitoring machinery health, operational efficiency, and diagnosing faults. As advancements in electronics, computer technologies, and manufacturing processes surge, so do the fields of vibration sensing and measurement. This evolution has brought about a diversification and refinement in measurement methodologies and sensor technologies [1–3]. Optical-laser-based techniques have gained significant recognition and interest due to their ability to deliver comprehensive field information non-intrusively. Characterized by their high resolution and precision, these methods offer rapid and robust solutions in various applications [4–6]. These optical measurement techniques, preserve the dynamic characteristics of the vibrating object, as they do not impose additional mass or forces upon it. Consequently, several non-destructive testing and evaluation tools have emerged, solidifying their status as effective and reliable methods. Key among these is various forms of optical metrology, such as interferometry [7,8], shearography [9], speckle interferometry [10], and holographic interferometry [3,11,12], each offering unique advantages in the realm of vibration analysis and materials testing.

Orbital Angular Momentum (OAM), defined mathematically as $exp(il\theta)$ where $\theta$ is the azimuthal angle and $l$ is the Topological Charge (TC), has been identified as a distinct physical dimension with applications in various optical technologies. These include holography and optical multiplexed displays, as evidenced in recent studies [13,14], as

well as in areas like optical storage [15,16], optical communication [17,18], and optical sensor [19,20], among others.

In optical sensors, OAM has several advantages. In our previous work [19], an OAM sensor was used to simultaneously measure the dynamic micro-displacement and the direction of a moving object in real-time. Y. Ren and colleagues successfully achieved simultaneous and independent detection of both rotational and linear motion. This was accomplished through the implementation of dual interference involving two conjugated OAM beams and a Gaussian beam [21]. Kerschbaumer et al. proposed that an OAM interferometer be applied to directly measure the concentration of NaCl and glucose solutions in a label-free and situ manner and to monitor temperature differences in the millikelvin to microkelvin range [22]. Z. Zhang et al. studied the tiny velocity detection method based on a rotating petal-shaped light structure and single-point detector and obtained a measurement error rate of less than 10 nm/s [23]. The principal mechanism of these measurement methods involves generating petal-shaped light patterns through the interference of two conjugated OAM beams (OAM-OAM). The motion of these petal-shaped patterns is then used to obtain information about the movement of the measurement target. However, The approach of utilizing interference between Gaussian and OAM (GS-OAM) light for target measurement has been relatively less discussed in the literature.

In this paper, we propose a dynamic micro-vibration measurement based on OAM. In this method, Gaussian light serves as the reference beam and OAM light as the measurement beam. The interference between these two beams also generates petal-shaped light patterns, with the number of petals being equal to $|l|$. The OAM light reflected from the target carries information about the target's vibrations. The interference of the OAM light with Gaussian light forms petal-shaped patterns and the target's vibration information is measured through the rotation angle of these petals. Then, the intensity of the petals at a fixed radius is mapped onto Cartesian coordinates. The rotation angle of the petals is measured using the Fast Fourier Transform (FFT) method.

Compared to the OAM-OAM method [19,21–23], our proposed GS-OAM method offers two main advantages. Firstly, the combination of Gaussian and OAM light more readily produces petal-shaped light patterns, and misalignment has a significantly lesser impact on the GS-OAM method than on the OAM-OAM method. Secondly, the number of petals produced by our method is $|l|$, in contrast to the $2|l|$ petals generated by the OAM-OAM method. As a result, our proposed method not only has a higher measurement resolution but also enhances the accuracy of measurements.

## 2. Materials and Methods

### 2.1. The Principle of OAM-Vibration Measurement

The scheme of the proposed method is illustrated in Figure 1, where Figure 1a–c show the optical setup, the first frame result of the charge-coupled device (CCD), and the second frame result of CCD, respectively.

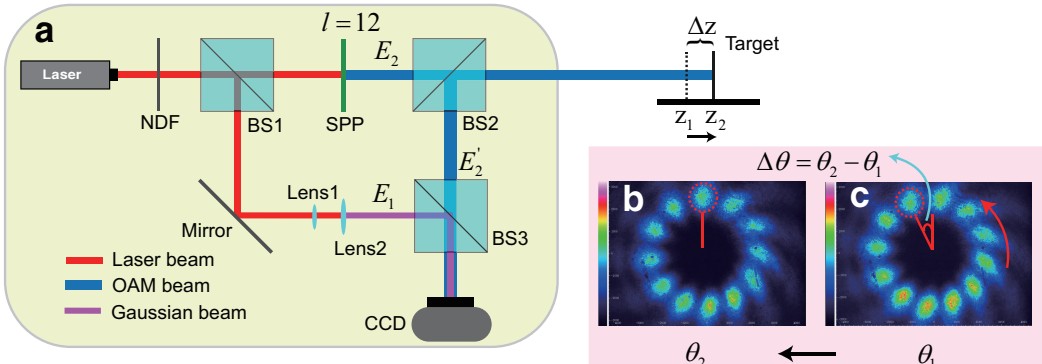

**Figure 1.** The scheme of the proposed method. (**a**) The optical setup. (**b**) The first frame result of CCD. (**c**) The second frame result of CCD.

In the optical setup illustrated in Figure 1a, the laser beam is directed through a Neutral Density Filter (NDF), employed to modulate the laser beam's intensity. Then, the laser beam is split into two by Beam Splitter 1 (BS1): one serving as the reference beam and the other as the measurement beam. The reference beam is angle-adjusted using a mirror, enabling it to interfere coaxially with the measurement beam. Subsequently, the beams pass through Lens 1 and Lens 2, which expand their spread, effectively transforming them into the Gaussian beam. Lastly, the reference beam can be described as

$$E_1 = A_1 exp(i\phi), \tag{1}$$

where the $A_1$ and $\phi$ represent the amplitude and phase of the Gaussian beam.

On the other hand, the measurement beam, upon passing through the Spiral Phase Plate (SPP), generates an OAM beam. The OAM beam can be given by

$$E_2 = A_2 exp(il\theta), \tag{2}$$

where the $A_2(l, r) = l^{2|l|}e^{-2r^2}$ represents the amplitude the OAM beam. $r$ represents the radius in polar coordinates. The OAM beam, after passing through Beam Splitter 2 (BS2), illuminates the vibrating target. The OAM beam, reflected back from the target, carries motion information about the target's vibration amplitude and direction. The light field of reflected OAM beam ($E_2'$) can be described as

$$E_2' = A_2' exp(il\theta)exp(ik2z), \tag{3}$$

where $z$ represents the distance between the target and BS2, $k = 2\pi/\lambda$ denotes the wave number, and $\lambda$ is the wavelength of the measurement beam. $A_2'$ represents the amplitude the reflected OAM beam.

The reference and measurement beams achieve interference at Beam Splitter 3 (BS3), with the interference pattern captured by a CCD camera (see Appendix A), and the result is

$$\begin{aligned} I(\theta, r) &= (E_1 + E_2')(E_1 + E_2')^* \\ &= A_1^2 + A_2'^2 + 2A_1 A_2' cos[l\theta + (2kz - \phi)]. \end{aligned} \tag{4}$$

From the Equation (4), the intensity of the petal-shaped is modulated as $2A_1 A_2' cos[l\theta + (2kz - \phi)]$, where $|l|$ denotes the number of petals. These petals undergo rotation by an angle of $kz$, which are shown in Figure 1b,c. To precisely measure the rotation angle of the petals, the maximum intensity value ($I_{max}$) is identified and used as the feature point, representing the brightest position on each petal. $I_{max}$ can be obtained, when $l\theta + (2kz - \phi) = 0$ from Equation (4). Hence, in the $k$th frame, the relationship between the petals' angle ($\theta_k$) and the distance($z_k$) is depicted as

$$\theta_k = -\frac{2\pi z_k}{\lambda \ell} + \phi, \tag{5}$$

where $k$ indicates the $k$th frame captured by the CCD camera, whereas $\phi$ represents the constant phase of the Gaussian beam. The relationship between the time ($t$) and the frame ($k$) can be described as

$$t = fps \cdot k, \tag{6}$$

where $fps$ represents the frame per second of CCD. Lastly, The relationship between the vibration amplitude ($\Delta z_k$) and the rotation angle of the petals ($\Delta \theta_k$) in CCD can be expressed as

$$\begin{aligned} \Delta\theta_k &= \theta_k - \theta_1 \pm 2\pi m \\ \Delta z_k &= z_k - z_1 = -\frac{\lambda l}{4\pi}\Delta\theta_k, \end{aligned} \tag{7}$$

where $\theta_1$ denotes the petals' initial angle, while $z_1$ signifies the target's initial distance. $\theta_k$ represents the petal angle of the k-th frame, ranging from 0 to $2\pi$. When the petals continuously rotate clockwise by more than $2\pi$, the rotation angle is adjusted by adding $2\pi m$, where $m$ represents the number of rotations. On the other hand, when the petals continuously rotate counterclockwise by more than $2\pi$, the rotation angle is adjusted by subtracting $2\pi m$. The rotation direction can be determined by subtracting the results of adjacent frames ($\Delta\theta_k - \Delta\theta_{k-1}$). A positive result indicates clockwise rotation, while a negative result indicates counterclockwise rotation.

When the TC is positive and the target moves toward BS2, the petals on the CCD rotate clockwise. Conversely, when the target moves away from BS2, the CCD petals rotate counterclockwise. On the other hand, when TC is negative and the target approaches BS2, the CCD petals rotate counterclockwise. As the target recedes from BS2, the petals rotate clockwise. As an illustration, consider the target's initial position at $z_1$, with the corresponding petal angle as shown in Figure 1b. In the second frame, when the target moves to $z_2$, The corresponding petals rotate counterclockwise, which is depicted in Figure 1c. The movement of the target by a distance of $\Delta z$ results in a rotation of the petals by an angle $\Delta\theta$.

### 2.2. The Method for Measuring the Petal's Angle

The method for measuring the petal's angle is shown in Figure 2. Firstly, the images captured by the CCD are converted into grayscale. The grayscale image of the first frame is shown in Figure 2a. From Equation (4), the interference intensity result comprises two components: the direct current (DC) component and the beat signals. Disregarding the influence of the direct current (DC) component on the measurement and selecting a measurement radius ($r_0$, blue circle in Figure 2a), the result can be represented as

$$I(\theta, r_0) = 2A_1 A_2' cos[l\theta + (2kz - \phi)], \tag{8}$$

where $I(\theta, r_0)$ is a cosine signal. The results obtained from sampling $\theta$ are transformed from polar coordinates to Cartesian coordinates, and are represented as

$$I(\theta(n)) = 2A_1 A_2' cos[l\theta(n) + (2kz - \phi)], \tag{9}$$

where $\theta(n) = \dfrac{2\pi}{N}n$ is the sampling $\theta$. The range of $\theta$ is from 0 to $2\pi$ and the sampling interval is $(2\pi/N)$, where $N$ is the total sampling number. In our experiment, the total sampling number is 1000 ($N = 1000$). The sampling intensity is shown in Figure 2b. The petal's position of the red circle in Figure 2a is mapped to the position of the red circle in Figure 2b. The sampled intensity in Cartesian coordinates is a cosine signal, similar to a communication signal. In Figure 2a, the green point denotes the starting point of the sampling, and the red point signifies the endpoint of the sampling. Concurrently, these points are mapped to the zero and final sampling points in Figure 2b. The orange line in Figure 2b is derived from the fitting of the blue line data points, representing a standard sine wave. Hence, we employ a method commonly used in communications to extract the phase information of the petals' angle, which can be described as

$$\theta_1 = angle\{FFT[I_1(\theta(n))]\}, \tag{10}$$

where $FFT[\cdot]$ denotes the application of the Fast Fourier Transform (FFT) to the signal. $angle\{\cdot\}$ represents the extraction of phase information from a complex signal.

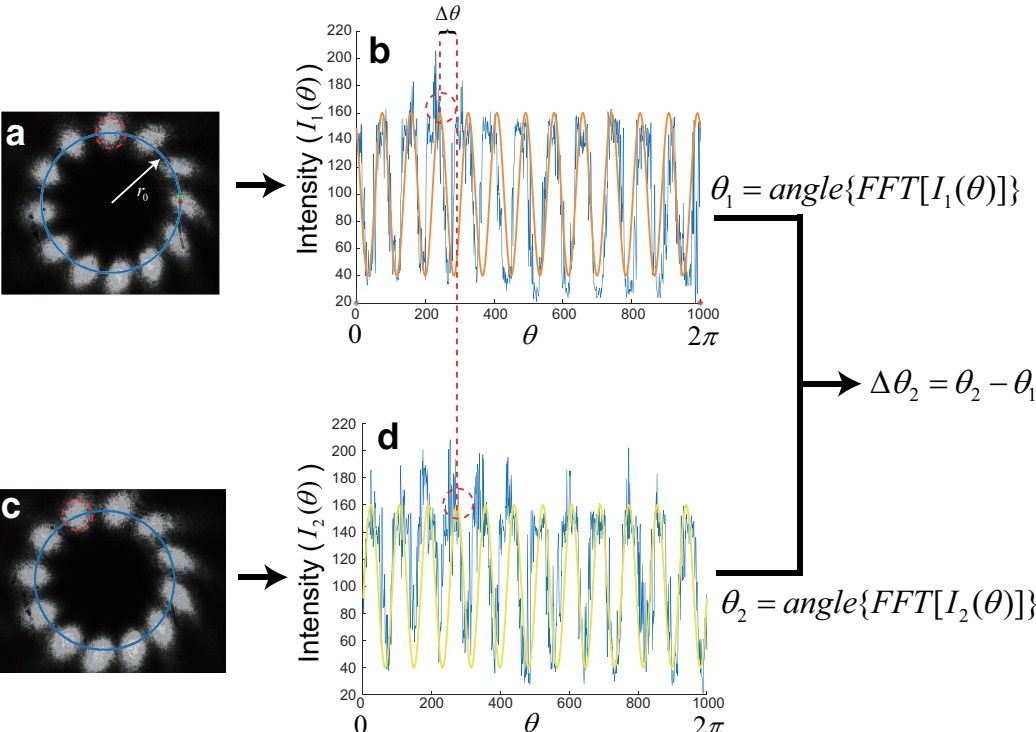

**Figure 2.** (**a**) The grayscale image of the first frame, where the blue circle represents the measurement circle with the measurement radius ($r_0$), where the green point denotes the starting point of the sampling, and the red point signifies the endpoint of the sampling. (**b**) The sampling intensity result, where the orange line represents the result of the fitting process, and the green and red points represent the corresponding green and red points in Figure 2a. (**c**) The grayscale image of the second frame, where the blue circle represents the measurement circle with the measurement radius ($r_0$). (**d**) The sampling intensity result, where the yellow line represents the result of the fitting process.

In the second frame, the grayscale image of the second frame is shown in Figure 2c. The red circle in Figure 2a rotates to the position of the red circle in Figure 2c. Notably, the measurement radius is also $r_0$, and the sampled intensity information is illustrated in Figure 2d. The petal's position of the red circle in Figure 2c is mapped to the position of the red circle in Figure 2d. The yellow line in Figure 2d is based on the fitting results of the sampled signals, and it too represents a standard sine wave. Following the same procedure, the phase information of the petals' angle can be extracted, which can be described as

$$\theta_2 = angle\{FFT[I_2(\theta(n))]\}. \tag{11}$$

Finally, the angle of rotation is given by

$$\Delta\theta_2 = \theta_2 - \theta_1. \tag{12}$$

*2.3. Optical Setup*

In order to validate the efficacy of the proposed method for OAM vibration measurement, a proof-of-principle experiment is conducted. The optical setup is illustrated in Figure 3.

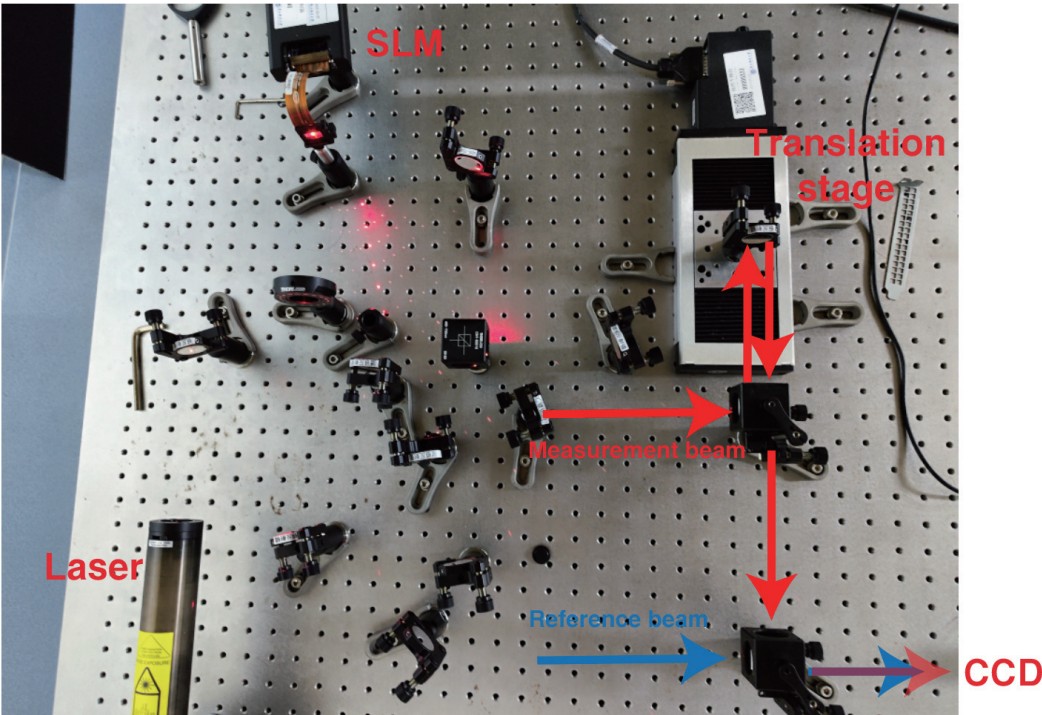

**Figure 3.** The photograph of the experimental setup.

A diode-pumped solid-state laser (MGL-III-532-100mW) generates a laser beam with a wavelength of 632.8 nm. The OAM beam is produced using a spatial light modulator (SLM, Holoeye PLUTO-2-VIS), featuring 1920 × 1080 resolution, a pixel pitch of 8.0 μm, an active area of 15.36 × 8.64 mm², and a 60 Hz refresh rate. In our experiment, we substitute the SPP with an SLM, which facilitates the convenient adjustment of TC. Micro-vibration is achieved using a high-performance precision translation stage (Newport-UTS-CC), which includes a linear encoder for movement verification. The CCD camera operates at a frame rate of 30 frames per second (30 fps). For convenience, the radian system is converted to the angle system ($2\pi \to 360°$) and the relationship between the vibration amplitude ($\Delta z_k$) and the rotation angle of the petals ($\Delta\theta_k$) can be expressed as

$$\Delta z_k = -\frac{\lambda l}{2 \cdot 360°} \Delta\theta_k. \tag{13}$$

When the rotation angle of the petals is 1° ($\Delta\theta = 1°$) with a wavelength of 632.8 nm ($\lambda$ = 632.8 nm), the vibration amplitude is $\Delta z$ = 0.88 l (nm), and the measurement resolutions of OAM-OAM and GS-OAM corresponding to different TCs ($l$) are presented in Table 1.

**Table 1.** The measurement resolutions with different TCs ($l$).

| TC | $l = 2$ | $l = 4$ | $l = 6$ | $l = 8$ | $l = 10$ |
|---|---|---|---|---|---|
| OAM-OAM | 3.52 nm | 7.04 nm | 10.56 nm | 14.08 nm | 35.20 nm |
| GS-OAM | 1.76 nm | 3.52 nm | 5.28 nm | 7.04 nm | 17.60 nm |

To quantitatively evaluate the experimental results, the average deviation (A.D.) is used, which is defined as

$$A.D. = \frac{1}{K} \sum_{i=1}^{K} \left| \Delta z(k) - \Delta z'(k) \right|, \tag{14}$$

where $\Delta z_k$ and $\Delta z'_k$ represent the measured and simulation vibration amplitude in the kth frame, respectively.

## 3. Results

The high-performance precision translation stage is utilized to simulate the target's vibration, with its amplitude set to vary from 20 nm to 160 nm and a vibration frequency of 2 Hz. TC of OAM is set to 2 ($l = 2$). The measurement results are presented in Figure 4. Figure 4a displays the dynamic vibration amplitude measurements over 5 s with $l = 2$. Figure 4b shows the CCD images at frames A, B, and C, along with the corresponding rotational angles and vibration amplitudes.

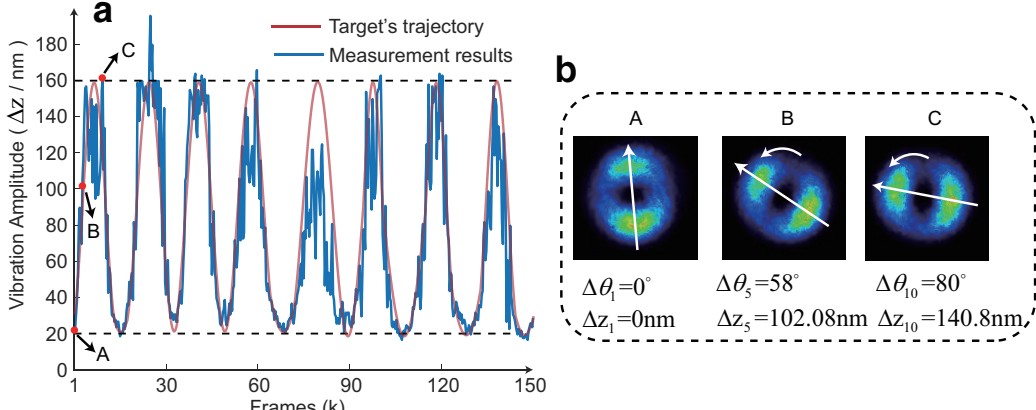

**Figure 4.** (**a**) The dynamic vibration amplitude measurements over a period of 5 s with $l = 2$, where the red line is the vibration trajectory of the target and the blue line is the measurement results. (**b**) The CCD images at frames A, B, and C, along with the corresponding rotational angles and vibration amplitudes.

In Figure 4a, the vibration amplitude is plotted on the y-axis in nanometers (nm), and the frames are on the x-axis. The red line is the vibration trajectory of the target and the blue line is the measurement results. The measurement results oscillate between 20 nm and 160 nm, indicating that the vibration amplitude fluctuates within this range. The measured vibration frequency is 1.8 Hz, which aligns with the experimental setup parameters. The Average Deviation (A.D.) of this measurement is A.D. = 22.34 nm. Therefore, our proposed measurement method enables dynamic high-precision micro-vibration measurement.

Figure 4b presents the CCD results for the 1st, 5th, and 10th frames from Figure 4a, with points A, B, and C corresponding to the marked positions respectively. The 1st frame represents the initial position with a rotation angle of 0° and a vibration amplitude of 0 nm. In the 5th frame, the petals are observed to rotate counterclockwise by 58°, indicating a motion of 102.08 nm away from the measurement system. By the 10th frame, the petals continue to rotate counterclockwise by 80°, corresponding to a continued movement of 140.8 nm away from the measurement system.

## 4. Discussion

### 4.1. Impact of Misalignment on Measurement Accuracy

The center distance ($d$) between the centers of the reference and measurement beams affects the quality of the petal formation. To assess the impact of misalignment on the measurement of vibration amplitude, different center distances are used to measure the vibration amplitude. The results of the A.D. are presented in Figure 5a.

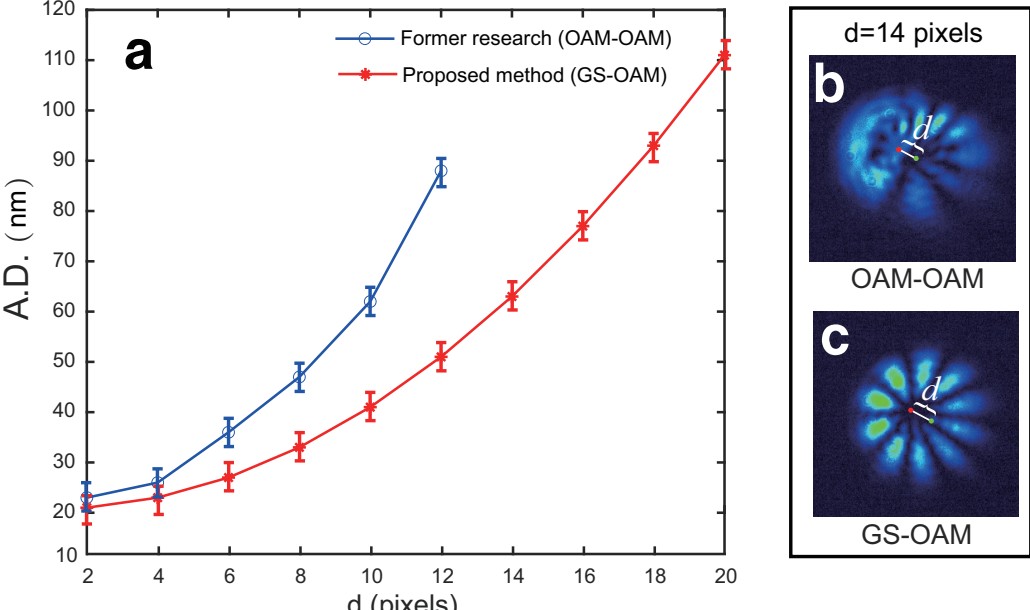

**Figure 5.** (**a**) The average deviation of our proposed OAM sensor and former research with different center distances. (**b**) The CCD image of OAM-OAM with *d* = 14 pixels. (**c**) The CCD image of GS-OAM with *d* = 14 pixels. Red point and green point represent the center of the reference and measurement beams.

In former research [19], both the reference and measurement beams are OAM beams (OAM-OAM). In contrast, our proposed method utilizes a Gaussian beam for the reference and an OAM beam for the measurement (GS-OAM). The experimental results show that both OAM-OAM and GS-OAM configurations are highly sensitive to alignment inaccuracies. The A.D. increases with the increasing center distance. Notably, this increase in A.D. is more pronounced in the OAM-OAM configuration. When the distance *d* is 14 pixels (*d* = 14 pixels), the OAM-OAM fails to perform normal measurements, with the CCD results shown in Figure 5b, where the red point and green point represent the center of the reference and measurement beams, respectively. In contrast, the GS-OAM configuration can still perform normal measurements. The CCD results for this setup are depicted in Figure 5c, where the red point and green point represent the center of the reference and measurement beams, respectively. As observed in Figure 5c, the petals exhibit noticeable unevenness, which is a primary factor affecting the measurement accuracy. Therefore, our method offers a higher tolerance for misalignment.

### 4.2. Effect of Vibrational Frequency on Measurement Accuracy

Our proposed method and the laser doppler vibrometer (LDV) method [24] measure the same target with different vibrational frequency. The impact of vibrational frequency on measurement accuracy are shown in Figure 6. At lower vibration frequencies, the measurement error of the LDV is notably greater than that of our proposed method. However, as the vibration frequency increases, the A.D. of our method increases sharply. In contrast, the measurement error of the LDV method increases more gradually with the vibration frequency when compared to our proposed method. Therefore, our method demonstrates an advantage in measuring slow vibrations.

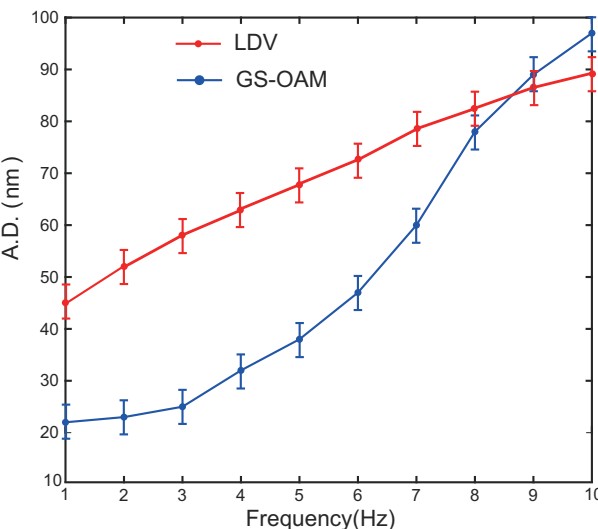

**Figure 6.** The average deviation of measurement results with different vibrational frequencies (Hz), where the red line represnets the measurement results of the laser doppler vibrometer (LDV) and the blue line represents the measurement results of the proposed method.

*4.3. Effect of CCD's fps on Measurement Accuracy*

The relationship between the average deviation (A.D.) of measurement results and the frame rate (fps) of the CCD camera is shown in Figure 7. The experimental result indicates a decrease in A.D. as the fps increases, suggesting that higher frame rates lead to more accurate measurements. Specifically, the A.D. starts at a higher value when the fps is low, at around 5 fps, and steadily diminishes, reaching its lowest at 30 fps.

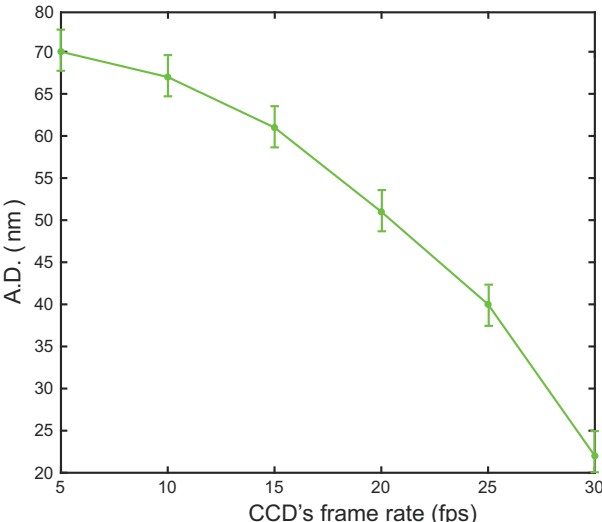

**Figure 7.** The average deviation of measurement results with different frame rates (fps).

The primary reason for the observed trend is that a higher fps of the CCD captures more detailed information on the rotation of the petals. Consequently, measurement precision can be enhanced by employing a high-speed CCD.

*4.4. Effect of TC on Measurement Accuracy*

Different TCs are employed to measure the same target, and the measurement results are illustrated in Figure 8. Both the average deviation (A.D.) from previous research and the proposed method increase with the rise of the TC. According to Equation (7), the measurement resolution varies with different TCs. The same angular measurement error of 1° signifies different levels of accuracy for varying TCs.

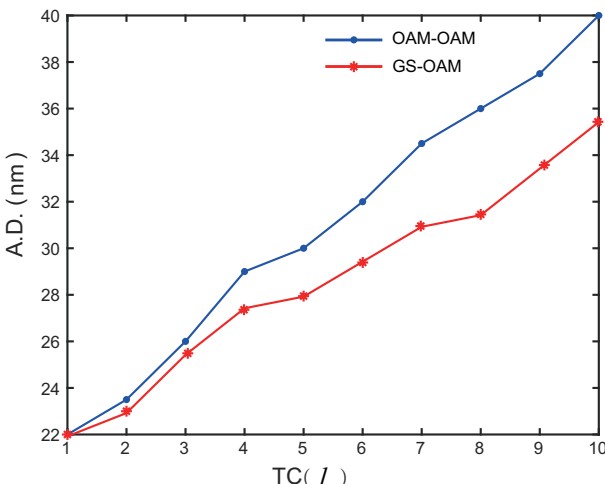

**Figure 8.** The average deviation of measurement results with different TCs (*l*).

However, the rate of increase in A.D. for our proposed method is significantly lower compared to that of the OAM-OAM. This is primarily because our angle measurement technique, which employs the Fast Fourier Transform (FFT), is more precise, effectively suppressing the rise in A.D.

*4.5. Measurement Range*

The measurement range for configurations where both the reference and measurement beams are OAM (OAM-OAM), and where the reference beam is Gaussian and the measurement beam is OAM (GS-OAM), is depicted in Figure 9. The range for OAM-OAM increases with the Topological Charge (TC) because the size of the measurement beam's rings expands with the path length, leading to a mismatch in ring size compared to that of the reference beam, which prevents interference.

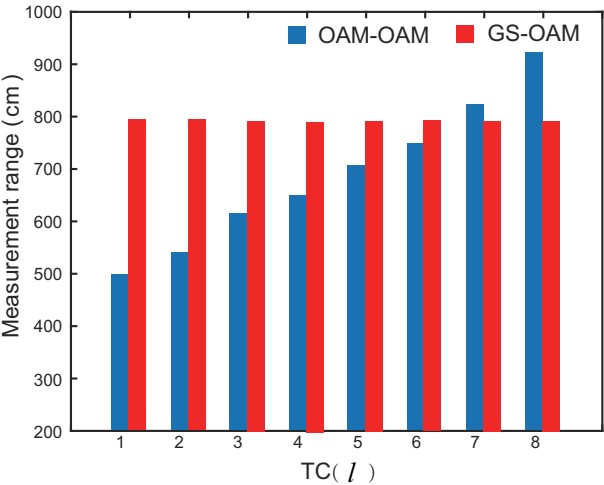

**Figure 9.** The measurement range of OAM-OAM and GS-OAM.

In contrast, GS-OAM does not face this issue because the Gaussian of the reference beam allows for easy adjustment of the beam size using expanding optics (Lens1 and Lens2). However, the limiting factor for GS-OAM's measurement range is the mismatch in light field intensity. As the distance increases, the intensity of the OAM measurement beam weakens, while the Gaussian reference beam remains strong, overwhelming the measurement beam. Therefore, the measurement range for our proposed method extends from 0 cm to 800 cm (0–800 cm).

*4.6. The Advantages and Disadvantages of the Proposed Method*

In the realm of dynamic micro-vibration measurement, the GS-OAM method marks a significant advancement, particularly in comparison to the traditional OAM-OAM approach. One of its most notable advantages is its enhanced tolerance to misalignment, which ensures stable and reliable measurements even when perfect alignment is not achievable. This characteristic is crucial for maintaining consistency in results under less-than-ideal conditions. Moreover, the GS-OAM method excels in measurement accuracy, boasting a precision up to 22.34 nm. This level of accuracy is particularly important in fields that require nanometer-scale resolution, ensuring that even the smallest vibrations are accurately captured.

Additionally, the GS-OAM method offers an extended measurement range of 0 to 800 centimeters. This wide range makes it adaptable for various applications, from short to long-distance measurements. Such versatility is especially beneficial in diverse fields that require different measurement scales. Lastly, the practicality of the GS-OAM method in a multitude of applications cannot be overstated. Its precision and range make it an ideal choice for applications such as precision engineering, high-resolution microscopy, and spacecraft stabilization systems. These applications benefit greatly from the method's ability to provide accurate measurements across a broad spectrum of conditions and requirements, demonstrating the GS-OAM method's broad applicability and effectiveness in dynamic micro-vibration measurement.

The GS-OAM method, while offering notable advantages in dynamic micro-vibration measurement, is not without its disadvantages. One primary limitation is the impact of the vibrational frequency on accuracy. As the target's vibrational frequency increases, the accuracy of measurements tends to decrease. This presents a challenge in scenarios involving high-frequency vibrations, where precision is paramount. Another critical factor influencing accuracy is the frame rate of the CCD camera used in the measurement process. Lower frame rates lead to a higher average deviation in measurements, implying a necessity for high-speed cameras to achieve optimal accuracy. This requirement can limit the method's applicability in situations where high-speed cameras are not available or feasible.

The method's accuracy is also affected by the topological charge (TC) of the beam. Different TCs can result in varying levels of measurement precision, necessitating careful calibration for each specific TC. This aspect adds complexity to the measurement process and can be a potential hurdle in ensuring consistent accuracy across different applications. Furthermore, the GS-OAM method faces challenges with intensity mismatches at longer distances. As the measurement distance increases, the intensity of the OAM measurement beam diminishes relative to the Gaussian reference beam, potentially impacting the accuracy of measurements over long ranges.

In summary, while the GS-OAM method provides significant benefits in terms of tolerance to misalignment, measurement accuracy, and range, it also possesses limitations that need consideration in practical applications. These include the effects of vibrational frequency, CCD frame rates, topological charges, and intensity mismatches, especially over long distances. Acknowledging and addressing these limitations is crucial for the effective utilization of the GS-OAM method in various dynamic micro-vibration measurement scenarios.

## 5. Conclusions

In conclusion, our study presents a robust approach to dynamic micro-vibration measurement by utilizing a configuration where the reference beam is Gaussian and the measurement beam is OAM (GS-OAM). This method showcases a higher tolerance for misalignment compared to the OAM-OAM configuration, as evidenced by the lower rate of A.D. increases with misalignment. The impact of vibration frequencies, CCD frame rates (fps), and the TCs on the measurement accuracy are analyzed. The experimental results show that our proposed method has a higher accuracy 22.34 nm and a longer range (0–800 cm). These attributes make our proposed method superior in terms of

precision and practicality for applications requiring nanometer-scale resolution in vibration measurements, such as in precision engineering, high-resolution microscopy, and spacecraft stabilization systems.

In future work, the concept of utilizing dynamic fields offers an intriguing prospect for enhancing the capabilities of the GS-OAM method [25,26]. The integration of dynamic field modulation into the OAM framework promises to bring about substantial improvements in terms of precision and control. This enhanced flexibility could not only refine the existing functionalities of the GS-OAM method but also potentially introduce new ones. The ability to dynamically modulate OAM paves the way for more adaptable and responsive measurement techniques, which could be particularly advantageous in rapidly changing or unpredictable experimental environments. This exploration into dynamic field modulation represents a significant step towards realizing the full potential of OAM in practical and experimental scenarios, and we are enthusiastic about pursuing this promising avenue in our future research endeavors.

**Author Contributions:** Conceptualization, J.Z. and F.Z.; methodology, J.Z. and L.W.; software, J.Z. and L.W.; validation, L.W.; writing—original draft preparation, J.Z. and F.Z.; writing—review and editing, J.Z., F.Z. and S.Z.; supervision, F.Z. and S.Z. All authors have read and agreed to the published version of the manuscript.

**Funding:** This research was funded by the National Natural Science Foundation of China (NSFC) (62375140), the Natural Science Foundation of Suqian (K202209), the Aba Teachers University Project (AS-PYYB2023-02), and the Open Research Fund of National Laboratory of Solid State Microstructures under Grant (36055).

**Institutional Review Board Statement:** Not applicable.

**Informed Consent Statement:** Not applicable.

**Data Availability Statement:** The data related to the paper are available from the corresponding authors upon reasonable request.

**Conflicts of Interest:** The authors declare no conflict of interest.

**Appendix A**

The result of the interference between Gaussian and OAM light as captured by the CCD can be described as

$$
\begin{aligned}
I(\theta, r) &= (E_1 + E_2^{'})(E_1 + E_2^{'})^* \\
&= [A_1 exp(i\phi) + A_2^{'} exp(il\theta) exp(i2kz)][+A_1 exp(-i\phi) + A_2^{'} exp(-il\theta) exp(-i2kz)] \\
&= A_1^2 + A_2^{'2} + A_1 A_2^{'} exp(il\theta + i2kz - i\phi) + A_1 A_2^{'} exp[-(il\theta + i2kz - i\phi)].
\end{aligned}
\tag{A1}
$$

According to Euler's theorem, the equation can be converted into a cosine function, which can be described as

$$
\begin{aligned}
I(\theta, r) &= (E_1 + E_2^{'})(E_1 + E_2^{'})^* \\
&= A_1^2 + A_2^{'2} + 2A_1 A_2^{'} cos[l\theta + (2kz - \phi)].
\end{aligned}
\tag{A2}
$$

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
