# Peer review of "Dynamic Micro-Vibration Measurement Based on Orbital Angular Momentum"

_photonics, doi:10.3390/photonics11010027_

Round 1
Reviewer 1 Report
Comments and Suggestions for Authors
In this manuscript,a scheme for dynamic micro-vibration measurement using an Orbital Angular Momentum (OAM) interferometer where the reference beam is Gaussian (GS) is proposed and experimentally demonstrated. The measurement performance is compared with that of OAM-OAM and show good superiority. However, there are some concerns about the measurement.
1) The micro-vibration is measured by the rotation of the petals’ angle. However, the angle has the period of 2pi, then how to differential the confusions between theta and theta+2npi?
2) In fig.3a, Does the vibration oscillate periodically? If so, the measurement error is not small. It is said that A.D=22.34nm and thus the measurement accuracy of this scheme. However A.D is related to the measured frames. Is this conclusion correct?
Comments on the Quality of English LanguageThere are some typos in the manuscript.
1)P.1, the last 6 line, there is a question behind {2,9?}
2)P.3, the 3 line below Eq.(4), “Figure 1b and Figure 1d”, there is no Figure 1d.
3) P.8, the last 3 line, “according to Equation ??”
Reviewer 2 Report
Comments and Suggestions for Authors
The manuscript ID photonics-2769911 mainly presents an original technique for dynamic micro-vibration measurement. The assistance of an Orbital Angular Momentum interferometer is employed and experimental results are analyzed. Please see below a list of comments to the authors.
1. The advantages and disadvantages of the system proposed should be described with better details.
2. The main results should be confronted with updated publications in the topic of micro-vibration measurement systems assisted by optical effects.
3. From Figure 1 seems that the measurement of the vibration is only proposed for unidimensional vibrations. Please edit the figure if the measurement can be for bidimensional or tridimensional phenomena.
4. In the manuscript is described: “These capabilities render our technique highly suitable for applications… including geophysical vibration monitoring” But geophysical vibration monitoring requires 3D measurements. Please support the asseveration within the text.
5. The Orbital Angular Momentum (OAM) is governed by a neutral density filter and a spiral phase plate in this work. The authors are invited to discuss some perspectives for future work considering the potential modulation of the OAM by using dynamic fields. You can see for instance: DOI: 10.1039/d1cp05195d and https://doi.org/10.1103/PhysRevApplied.19.014045
6. The authors wrote “our proposed method not only has a higher measurement resolution but also enhances the accuracy of measurements” It would be useful for readers the incorporation of a table showing the resolution reported in the methods compared.
7. Additional keywords can be added.
8. A photograph of the experimental setup would be welcome.
9. The characteristics of the laser source and components of the experimental setup should be described in the section of Materials and Methods instead of presenting them in the Results section.
10. Error bar in experimental irradiance data recorded must be reported.
Comments on the Quality of English LanguageA proofreading is suggested
Reviewer 3 Report
Comments and Suggestions for Authors
The authors present an approach for measuring dynamic micro-vibrations by utilizing a configuration where the reference beam is Gaussian and the measurement beam is orbital-angular-momentum (OAM) based.
The presented method allows for high tolerance for misalignment, as e.g. compared with the OAM-OAM configurations. The presented experimental results reveal an accuracy of around 22 nm, over a range of 0 − 800 cm.
These results might be useful for applications such as in geophysical vibration monitoring, precision engineering, high-resolution microscopy, etc.
From my perspective, the paper could be further considered in this journal after the authors address the following points:
1. The authors deploy a two-channel configuration. Though this optical scheme may be easy to build, the experimental setup appears to be complex, bulky, and expensive. Can these aspects be improved? Please comment.
2. As the two beams travel along different paths and through different optical components, aberrations may be introduced by these optical components, which may affect the measurement accuracy. Can this aspect be imrpoved? How is the accuracy limit infered herein?
3. Any slight mechanical vibrations, or small air disturbances in either of the two channels, may lead to lowering the spatial and temporal phase stability. Can this aspect herein be addressed? Please comment in some detail.
4. Important relevant previous works, such as: Measurement 15, 251 (1995); Appl. Opt. 33, 179 (1994); ACS Photonics 8, 296 (2020), etc, need to be cited in a revised version of the manuscript.
Round 2
Reviewer 2 Report
Comments and Suggestions for Authors
Besides the authors of the manuscript ID photonics-2769911 have clarified most of the points raised in the review stage, it seems that in the system is not present the updated reviewed version of the manuscript.
Some of the comments pointed out in the author’s reply are not addressed in the reviewed version of the manuscript. You can see for instance that the photo for response 8 is missing, and the discussion about future work described in point 5 is also missing in the manuscript.
Comments on the Quality of English LanguageA proofreading is suggested
